# Assessment of a Bionic Broach Implanted with Nylon Fibers

**DOI:** 10.3390/ma15249040

**Published:** 2022-12-17

**Authors:** Jing Ni, Haohan Zhang, Kai Feng, Huijun Zhao

**Affiliations:** School of Mechanical Engineering, Hangzhou Dianzi University, Hangzhou 310018, China

**Keywords:** bionics, flocked broach, mathematical model, broaching performance, 7075 aluminum alloy

## Abstract

The optimization of a broach surface is of great significance to improve the cutting performance of the tool. However, the traditional optimization method (surface texture, coating, etc.) destroys the stress distribution of the tool and reduces the service life of the tool. To avoid these problems, four kinds of flocking surfaces (FB1, FB2, FB3, and FB4), imitating the biological structure of Daphniphyllum calycinum Benth (DCB), were fabricated on the rake face of the broach by electrostatic flocking. The broaching experiment, wettability, and spreading experiment were then conducted. Moreover, the mathematical model of the friction coefficient of the bionic broach was built. The effect of broaches with different flocking surfaces on the broaching force, chip morphology, and surface quality of workpieces was studied. The results indicate that the flocked broaches (FB) with good lubricity and capacity of microchips removal (CMR) present a smaller cutting force (*F_c_*) and positive pressure (*F_t_*) compared to the unflocked broach (NB), and reduce the friction coefficient (COF). The chip curl was decreased, and the shear angle was increased by FB, which were attributed to the function of absorbing lubricant, storing, and sweeping microchips. Its vibration suppression effect enhanced the stability in the broaching process and improved the surface quality of the workpiece. More importantly, the FB2 with the most reasonable fluff area and spacing exhibited the best cutting performance. The experimental conclusions and methods of this paper can provide a new research idea for functional structure tools.

## 1. Introduction

Given their high production efficiency and wide processing range, broach tools have been employed in broaching metallic material. However, due to its higher cutting force and the difficulty of chip breakage, the surface quality of the workpiece deteriorates, and is unable to meet the high requirements for broaching performance, especially in the broaching of aerospace materials [1,2]. Furthermore, intense friction is produced between the rake face and the chip during the broaching of viscous materials (aluminum alloy, titanium alloy, etc.), resulting in a large amount of processing heat being generated. This heat is concentrated on the tool-chip interface, which leads to large deformation of the workpiece material [3]. In recent years, many researchers have tried to change the morphology of the rake face to reduce the friction of the tool–chip interface [4]. Therefore, it is of great significance to study the functional structure of broach surfaces [5]. At the same time, in the broaching of difficult-to-machine materials, such as aviation aluminum alloys, it is urgent to prepare a new broach to ensure lower cutting force and easy chip breaking, and improve the surface roughness of the workpiece.

At present, the main method to improve the cutting performance is to prepare a texture on the broach rake face [4,5,6,7,8]. Hao et al. [9] proposed a method for improving cutting tool anti-friction performance by constructing textured surfaces. In order to change tribological characteristics, Siju et al. [10] also fabricated innovative dual-texture geometries on the tool. Ashwani et al. [11] created tools with an optimal intermittent ratio and varying texture dimensions. Although textures can improve tool-cutting performance, they also destroy the stress distribution on the tool surface and reduce tool strength.

In addition, many scholars improved the processing of materials with coated tools [12,13,14,15]. Zhou et al. [16] investigated the effect of Co and cubic carbonitride content on the microstructure of alloy and cutting performance of coated tools. The experiment showed that the coating tools improved cutting performance by reducing flank wear. Lu et al. [17] fabricated the gradient TiAlSiN coating by magnetron sputtering to enhance the cutting performance of the titanium alloy. Xiong et al. [18] used a TiAlN-coated tool to study the machinability of TiB 2/7050Al composites. It was also found that the coating tool had a positive influence on the surface residual stress. However, a brittle decarburized layer is easily produced between the coating and the substrate as a result of the high coating temperature, leading to brittle fracture of the tool and reduced bending strength [19].

In the preparation of textured tools and coated tools, texture shape and coating materials often perplex many scholars. Therefore, some researchers draw inspiration from the perspective of bionics. Thus, in the processing of tool optimization, bionics were employed to solve some mechanical problems [20,21,22,23]. To test the bionic polycrystalline diamond compact drill, Wang et al. [24] created a bionic structure for the drill body and polycrystalline diamond compact cutter. Aiming at the problems listed above, Zhang et al. [25] combined the bionic structure with a ball-end milling cutter to study its cutting performance. Compared with the non-textured tool, the bionic ball-end milling cutter exhibited a stable cutting force and small fluctuation. Zheng et al. [26] optimized an example of bionic coupling and applied its shape to the surface of a workpiece. From the analysis of the mandible incisor profile of bamboo weevil larva, Tong et al. [27] found that the primary cutting edge of the incisor is close to a standard circular shape, which helps improve the cutting efficiency of the mandible incisor.

Mathematical modeling for material structures is also an effective way to optimize material behavior. By correlating the geometry of the functional structure of the tool surface with the cutting parameters of the tool, Amal et al. [28] created a model to assess the performance of a tool. Fabbrocino et al. [29] developed a three-dimensional model to study the wave dynamics of highly nonlinearly tensioned monolithic metamaterials. Lee et al. [30] applied the plasticity hypothesis of materials to solve processing issues. Fabbrocino et al. [31] proposed a novel method for the kinetic analysis of two-dimensional lattice material structures. Wenzel et al. [32] proposed a method for the evaluation of the wetting behavior of material surfaces. Pei et al. [33] also conducted a simulation study of the wetting of hair surfaces with different topologies. Therefore, modeling is widely used to assist in guiding the dynamics of materials.

It can be seen from the studies above that many scholars have established functional surfaces on the tool surface, such as texture technology or coating technology, to improve the mechanical performance of the tool [12,13,14,15,16,17,18,19,20,21,22,23,24,25,26,27]. However, these methods destroyed the stress distribution on the tool surface, which resulted in a reduction of the strength of the tool. In this study, fluff was implanted on the broach surface by electrostatic flocking technology to improve the properties of broaching 7075 aluminum alloy. Moreover, to study the relationship between the parameters of the functional surface and the processing performance, a mathematical model was established relying on the parameters of the flocked surface and machining performance. In addition, a broaching experiment was conducted to validate the flocking broach and mathematical model. The cutting force, chip morphology, and surface quality of workpieces were analyzed based on the mathematical model and experimental results.

## 2. Experiment and Method

### 2.1. Bionics Principle 

Different from other plants, the fluff of Daphniphyllum Calycinum Benth (DCB) (purchased from Anqing plant specimen Co., Ltd., Anqing, China) has an excellent ability to handle water and dust, which is consistent with the situation in the broaching process. To study the lipophilic properties of DCB, the surface morphology of DCB was taken by using a high-speed digital camera (Type: KEYENCE VW-9000, purchased from KEYENCE Co., Ltd., Shanghai, China) with a magnification of 500 times. The geometric structure of the DCB leaf surface is exhibited in Figure 1. The study found that the back micromorphology of DCB leaves has better lipophilicity than that of the surface of FC leaves, which is attributed to the villi-like microstructure on the DCB surface. Therefore, DCB was selected as the bionic prototype and applied to the rake face of the broach to study its broaching performance in the processing of aviation materials.

### 2.2. The Selection of Broach and Workpiece

The experimental cutting tool is a single-tooth broach (purchased from Wuhu Cemented Carbide Cutting Tools Co., Ltd., Wuhan, China), which is mainly used for the processing of turbine mortises. Its material composition is cemented carbide (YG8, Co = 8 wt% and WC = 92 wt%). It has significant advantages as the material of the tool, owing to high strength and wear resistance. The specific mechanical properties of the YG8 are listed in Table 1.

A 7075 aluminum alloy (purchased from Wuhu Cemented Carbide Cutting Tools Co., Ltd.) with a size of 10 × 20 × 30 mm was selected as the workpiece. Before the broaching experiment, the aluminum alloy’s surface was preliminarily treated with 2000 mesh sandpaper and 5000 mesh sandpaper, subsequently, and then cleaned by ultrasonically for 20 min. After being placed in a dry environment for one day, the broaching experiment was carried out on these workpieces. The specific parameters of the material are presented in Table 2 and Table 3.

### 2.3. The Preparation of the Flocked Broach

The preparation of flocking broaches was realized by electrostatic flocking technology, whose principle is exhibited in Figure 2a. The specific preparation process was as follows: in step one, since nylon only adheres to the adhesive, the area without fluff was covered by a rubber belt. In step two, the uncovered areas were painted with the adhesive to guarantee that the fluff was implanted in the desired position. In step three, the treated broaches were put into the electrostatic flocking box. In step four, the fluff was neatly arranged on the surface of the processed tools under the action of a magnetic field after being charged. And finally, the fluff was fixed on the rake face after the adhesive was dried and solidified. The broach tools were prepared in this study, as shown in Figure 2b.

The composition of the adhesive material was acrylic acid and polyurethane (purchased from Hangzhou Yunmao Plastic Co., Ltd., Hangzhou, China), and the material of fluff was white Nylon (purchased from Hangzhou Yunmao Plastic Co., Ltd.). Because a fiber of 0.8 mm has good wettability [34], a fiber of this length is used in this experiment. The specific electrostatic flocking parameters are shown in Table 4. In addition, Table 5 presents the area and spacing of five kinds of broaches (FB1, FB2, FB3, FB4, and NB).

### 2.4. Broaching Experimental Settings

The purpose of the experiment was to evaluate the changes in the broaching load and chip morphology, and the quality of the workpiece when using different bionic functional broaches. The comparative experiment of broaching with the flocked broaches and an unflocked broach was carried out on a single-tooth broaching testbed, and its layout is presented in Figure 3. The specific broaching parameters are shown in Table 6.

According to the experimental scheme, the four kinds of flocked broaches and the unflocked broaches were used to perform the broaching experiment. In the broaching process, the three-dimensional force sensor (type: KISTLER9119AA2, purchased from Jiangsu Donghua Test Technology Co., Ltd., Taizhou, China), with its acquisition frequency of 1024 Hz, was used to collect the broaching load. After broaching, the morphologies of the chip and the workpiece were observed by using a high-speed digital camera (type: KEYENCE VW-9000).

### 2.5. Wettability and Spreading Experiment

The wettability experiment was carried out to evaluate the lipophilicity of the flocked broach. Castor oil has a better lubrication effect compared with other lubricants. Therefore, castor oil was used for the wettability experiment, and an amount of 3 μL was dropped on the surface of the broach in each experiment. The contact angles of the droplets on the metal surface were collected every second by a contact angle measuring instrument (JC2000D1 purchased from Shanghai Zhongchen Digital Technology Equipment Co., Ltd., Shanghai, China).

Next, the spreading experiment of lubricant on the metal surface was implemented. A total of 3 μL of castor oil was dropped on the metal surface, and the spreading effect of the droplet was observed by a high-speed camera (Type: KEYENCE VW-9000), with the pictures taken every 1 s. Subsequently, the collected images were processed as follows: firstly, the images of droplet spreading were grayed by software, and then the contour of the droplet spreading was captured by adjusting the color threshold. Finally, the size was calibrated to measure the edge area of the processing image to obtain the spreading area of the droplet.

## 3. Modeling of the Friction Coefficient of the Bionic Broach

By establishing the mathematical model of the flocked broach, we can better understand the change of the broach loads in the process of broaching, such as the friction coefficient, shear angle, and so on. Therefore, based on the traditional mathematical model [30], the mathematical model of the flocked broach was established.

The relationship between the forces during processing is shown in Figure 4. However, only the main $F_{C}$ and $F_{t}$ can be collected by a three-dimensional dynamometer. Therefore, other forces can be associated with the $F_{C}$ and $F_{t}$, and the following formula was obtained:(1)$$F_{z}=\frac{F_{C}}{cos\left(\beta -\gamma \right)}\;$$
where $F_{z}$ is the resultant force of the positive pressure ($F_{t}$) and the cutting force ($F_{C}$);

*β* is the friction angle of the tool-chip interface; and *γ* is the rake angle of the tool.

Since the chip moves uniformly when the cutting state is stable, the resultant forces at the tool-workplace ($F_{z}$) and the tool–chip interface ($F_{R}$) are a pair of equilibrium forces. According to the force principle, $F_{R}$ can be obtained.
(2)$$F_{r}=F_{z}$$

Therefore, we can associate the force of the tool–chip interface with the force between the tool and the workpiece. The positive pressure force ($F_{n}$) between the rake face and the chip can be written as:(3)$$F_{n}=F_{r}cos\beta$$

The friction force can be expressed as the product of chip shear stress and shear surface area.
(4)$$F_{f}=A_{c}\tau_{s}$$
where $F_{f}$ is the friction force between the rake face and the chip, $A_{c}$ is the contact area of the shear surface, and $\tau_{s}$ is the shear stress.

In this study, when the nylon fibers were added to the rake face of the broach, the shear contact area was affected by the fluff, which can be expressed as follows:(5)$$A_{c}=l_{c}\cdot \left(\rho \overset{n}{\underset{i=1}{\sum }}w_{i}+\varepsilon \overset{n}{\underset{i=1}{\sum }}s_{i}\right)$$

Where $w_{i}$ represents the coating width; $s_{i}$ represents the width of the blank area, which is exhibited in Figure 2b; ρ and ε are the coating correction factor; and $l_{c}$ is the effective contact length between the chip and the broach tooth.

The effective contact length between the chip and broach tooth is affected by the chip curl angle and chip radius, so it can be expressed as follows:(6)$$l_{c}=l\cdot \theta \cdot R$$
where $l_{c}$ is the length of the fluff, *θ* is the chip curl angle, and *R* is the chip curling radius.

At the same time, when the fluff is added to the rake face of the broach tooth, the shear stress will also be affected by the fluff, which can be expressed as follows
(7)$$\tau_{s}=\;\xi \tau_{s0}$$
where ξ is the oil film thickness and $\tau_{s0}$ represents the original shear stress without the fluff coating.

In the broaching process, lubricant is usually used to improve the broaching performance. As presented in Figure 4, when the fluff is added to the surface of broach teeth, the lubricant remains on the surface of the broach for a longer time, and the oil film thickness can be expressed as follows:(8)$$\xi =\alpha \;h^{o}d^{p}e^{q}$$
where ξ is the oil film thickness, *h* is the fluff length, *d* represents the fluff diameter (approximately 8 µm), *e* is the spacing of fluff pattern, *o*, *p*, and *q* are the coefficients of the power function, and the adsorption coefficient α is a material-related factor.

The length of the fluff is an average value, which is calculated by intercepting the length of the fluff in a certain area.
(9)$$h=\frac{\sum_{i=1}^{n}h_{i}}{n}$$
where, *h_i_* is the length of the selected area.

The spacing *e* of fluff is an average value, which is calculated by intercepting the spacing of the fluff pattern in a certain area.
(10)$$e=\frac{\sum_{i=1}^{n\;}e_{i}}{n}$$
where $e_{i}$ is the spacing of the selected area.

The friction coefficient is a common evaluation parameter in the machining process, which is expressed as follows:(11)$$\mu \;\frac{\;\;F_{f}\;}{F_{n}}$$

When there is a stable oil film between the chip and the rake face, the friction coefficient and friction resistance can be effectively reduced when the chip slides on the rake face. By combining Equation (11), *μ* was obtained, which is expressed as follows:(12)$$\mu =\frac{l_{c}\cdot \left(\gamma \sum_{i=1}^{n}w_{i}+\sum_{i=1}^{n\;}s_{i}\right)\;\alpha h^{o}d^{p}e^{q}\xi \tau_{s0}\;\;\;\;\;\;\;\;\;\;\;\;\;\;\;\;}{F_{C}}$$

## 4. Results and Discussion

### 4.1. Influence of Tool Surfaces on the Broaching Force

#### 4.1.1. Broaching Force

Figure 5a–e shows the comparison of the cutting force between flocked broaches and the unflocked broach at different speeds. It was found that the cutting force (*F_c_*) of all broaches increased to varying degrees with the increase of the cutting speed. Compared with the NB, the flocked broaches presented a smaller cutting force. In addition, the cutting force of FB2 had the largest reduction. At the cutting speed of 20–60 mm/s, the cutting forces of FB2 were reduced by 16.52%, 16.81%, and 5.35% than that of the NB, respectively. As shown in Figure 5b, the broaching force of FB2 had a maximum reduction of 59.73 N, whereas the reduction of FB1, FB3, and FB4 were 16.25 N, 48.79 N, and 23.33 N, respectively.

The effect of the broach with different surfaces on positive pressure (*F*t) is illustrated in Figure 5d–f. The changing trends of positive pressure and cutting force are very similar under different broaching speeds. FB2 exhibits the best cutting performance due to the smallest cutting force. At the cutting speed of 20–60 mm/s, the positive pressure of FB2 reduces 23.95%, 24.41%, and 11.37%, respectively, in comparison to the NB. Nevertheless, it can be concluded that the flocking broaches can produce the best effect, as shown in Figure 5b. The processing performance of FB will reduce when the cutting speed is too high.

Due to the good lipophilicity of the flocked broaches, the friction of the tool–chip interface and the friction coefficient in the machining process can be reduced. Moreover, the lubricant can absorb the high temperature generated by friction and restrict the abrasive wear and the adhesive wear of the tool–chip interface caused by high temperature, thereby reducing *F_c_* and *F_t_* [32].

#### 4.1.2. Broaching Mechanism of Flocked Tools

The friction force generated in the broaching process is the main factor affecting the cutting force. Meanwhile, the coefficient of friction (COF) can well reflect the changes in *F_c_* and *F_t_* between different broaches. The relationship between the friction coefficient of broaches and the broaching forces can be calculated according to Equation (12).

To verify the effectiveness of the theoretical friction coefficient obtained by the mathematical model, the actual friction coefficient is calculated by the following formula.
(13)$$\mu =\frac{F_{t}+F_{c}tan\gamma }{F_{c}+F_{t}tan\gamma }$$
where *γ* is the rake angle; $F_{c}$ is the cutting force; and $F_{t}$ is positive pressure.

Figure 6 illustrates the relationship between the theoretical friction coefficient (FB1-t, FB2-t, FB3-t, FB4-t, and NB-t) and the actual friction coefficient (FB1, FB2, FB3, FB4, and NB) under different cutting speeds. It can be seen that, although the theoretical values are relatively larger than the experimental values, the errors are less than 5.6%, and the maximum error occurs at the 60 mm/s cutting speed of the FB4 broach. The reason may be that the uniform shear stress assumption and the correction coefficient cause small errors in the calculation, but the overall trend is highly consistent with the experimental value.

As shown in Figure 6, the theoretical friction coefficients of the flocked broaches are smaller than that of the unflocked broach, which also corresponds to the conclusions above. Taking the broaching speed at 40 mm/s as an example, FB2-t shows the best friction coefficient, and its coefficient of friction decreases by about 6.07%. FB3-t and FB4-t are reduced by 4.28% and 4.14%, respectively, compared with the traditional NB-t. At the same time, the reduction of FB1 is only 3.5%, which is the lowest in all FB-t broaches, indicating that the fluff area and spacing are too large or too small to achieve beneficial results. Based on the comparison, it is obvious that the FB2-t provides the best cutting performance, which is consistent with the conclusions above. Therefore, the results calculated by the mathematical model can well illustrate that all flocked broaches exhibit a better ability to suppress friction than that of the NB.

Cutting fluid plays an important role in lubrication and improving friction; therefore, the wettability experiment was carried out to evaluate the lipophilicity of FB and NB. As illustrated in Figure 7, the contact angles of two kinds of broaches are less than 90° at 1 s, indicating that all the broaches have good lipophilicity. In addition, the contact angle of the NB quickly drops to a relatively stable state within 1 s, but the contact angle of the FB continues to reduce. The reason for this phenomenon is that when the oil droplet contacts the surface of the fluff, the density of the fluff is quite large, which produces a thrust on the oil droplets. It follows the Cassie state and stays on the surface for a while. This is attributed to the existence of air cushions between the nylon and the oil droplet, which makes the oil droplets unable to enter the nylon smoothly [32]. After 1 s, the contact angle of the oil droplet on the surface of the FB broach decreases rapidly to zero. The reason for this phenomenon is that when the droplet enters the fluff, due to the capillary effect between the fluff, the droplet quickly diffuses between the fluff, which follows the Wenzel state, so that the contact angle of the droplet decreases rapidly [33].

The spreading experiment of castor oil on the rake face was implemented. The Original diagram (OD) and analysis diagram (AD) of oil spreading on the broach surface can be observed in Figure 8a. It can be seen that the droplets spread steadily on the rake face of the NB. However, the droplets spread rapidly on the rake face of the FB in an irregular manner. As presented in Figure 8b, the droplets spread rapidly on the NB surface within 1 s and slowly spread to a stable state after 1 s. In the meantime, for the FB, the droplets spread rapidly throughout due to Laplace pressure [33].

As shown in Figure 9a, the FB broach changes the contact area between the rake face and the chip and reduces the friction coefficient in the broaching process. More importantly, because the FB broach has good lipophilicity, it effectively ensures the lubricity and capacity of microchips removal (CMR) compared to NB, and thus reduces the *F_c_* and *F_t_*.

As presented in Figure 9c, the FB1 can only ensure good oil storage capacity. Therefore, after a period of broaching, since there is no fluff spacing, the tool surface produces excessive microchips. When microchips cannot be removed in time, the fluff of FB1 cannot sustainably support the microchips again, which will increase the contact stress of the microchips, thus improving the *F*_c_ and *F_t_* [26]. However, compared with FB1, FB3, and FB4, FB2 has a reasonable flocked area and spacing, which indicates that the fluff has a good adsorption capacity for lubricant, and sufficient lubricant is stored at the gap. Based on the thickness of the lubricant, FB2 also exhibits excellent CMR in the unflocked position. Therefore, FB2 presents the smallest *F_c_* and *F_t_*. For FB3 and FB4, since the flocking spacing is too large compared to FB2, the lubricity and the CMR cannot be guaranteed, resulting in the *F_c_* and *F_t_* being relatively larger.

In summary, it can be found that the lubricity of the tools increases with the increase of the flocking spacing, and the CMR of the tools increases with the increase of the flocked area, as shown in Figure 9b. Therefore, it can be concluded that there is a threshold value for the area and spacing of the flocked surface, and anything above or below this value will seriously affect the broaching force.

### 4.2. Influence of Different Tool Surfaces on Chip Morphology

#### 4.2.1. Chip Curl

The chip-breaking ability of the tool can be intuitively evaluated by the curling radius of the chip. Figure 10 presents the curling radius of the FB and NB broaches. It can be seen that the chip curling radius decreases with the increase in cutting speed. The reason is that the broaching temperature increases as the cutting speed increases, the bottom metal of the chip becomes soft, and the average friction coefficient and external friction force decrease. In addition, compared with the NB, the chip curling radius of the FB exist at different degrees of reduction, and FB2 shows the smallest curling radius in comparison to all broaches. At the speed of 40 mm/s, the curling radius of the NB is the largest, and the curling radius of the FB2 exhibits the maximum reduction, which is about 55%. The curling radius of the other flocked broaches is around 27.93–40.54% less than the NB.

Since the change of the shear angle can directly affect the chip curl, shear angle analysis can further explain the formation mechanism of the chip curl radius. Many researchers have proposed different shear surface theories, and the Lee–Scheff shear angle formula was quoted in by [30]. The authors believe that a large amount of heat was generated to soften the material and reduce the strength, which was attributed to the continuous large plastic deformation during cutting. Therefore, the shear angle was calculated based on the following formula combined with Equation (11).
(14)$$\varphi =\frac{\Pi }{4}+\beta -\gamma$$
where *β* is the friction angle, *φ* is the shear angle, and *γ* is the rake angle of the broach.

Figure 11 presents the relationship between the shear angle and FB and NB. Compared with the NB, the shear angle of the FB increases at different broaching speeds. At the broaching speed of 40 mm/s, although other flocked broaches also have obvious changes, the increase of FB2 is the most obvious, which increases by 6.29%. However, in comparison to the NB, the increase percentages of FB1, FB3, and FB4 were around 4.09%, 4.26%, and 4.69%, respectively. It is concluded that the shear effect on the material is different for different flocked broaches. Moreover, the chip curling leads to the reduction of the overall tool–chip contact length, correspondingly reducing the total tool–chip contact area. Consequently, the increase of the shear angle promotes an increase in the degree of chip curling to easily achieve chip breaking. The maximum increase in the shear angle of FB2 indicates that there is a minimum friction coefficient in the processing state, which is also consistent with the outcome of the discussion in Section 4.1.2.

#### 4.2.2. Chip Morphology

The chip is the spalling object of the workpiece under the action of the cutting tool, and the analysis of chip morphology can indirectly evaluate the surface quality of the workpiece. However, since the performance of all broaches is better at the cutting speed of 40 mm/s than that of other broaching conditions, only the results of this group are discussed. The shape of the chip is shown in Figure 12a at amplification of 150 times, which illustrates that the morphology of chip curling is easy to achieve for chip breaking. Moreover, there was no tearing or bur at the end of the chip, which indicates that the FB can efficiently suppress the cutting temperature [25]. However, as shown in Figure 12b, it can be found that the deep furrow caused by the NB is very obvious and serious. Compared with NB, the furrows produced by the FB are weakened. Among the FB1–FB4 broaches, the furrows of FB2 were the least, followed by FB3, and the furrows of FB1 and FB4 were similar. Therefore, FB2 displays the best cutting performance, followed by FB3, FB1, and FB4.

The reason for this phenomenon is the poor thermal conductivity of cemented carbide broaches, the high shear stress of the working materials, and the high adhesion of microchips. The microchips eventually adhere to the rake face to form a derivative cutting phenomenon. This kind of debris not only plays the role of chip breaking, but also produces friction in the broach–chip interface, resulting in furrows at the end of the chips [18,35].

The chip curling and chip shape mainly depend on the contact method of the broach surface. As shown in Figure 13, the flocked broaches can produce the following effects: firstly, compared with NB, which is presented in Figure 13a, the fluff itself has a better lipophilic characteristic, so it can produce better adsorption capacity for lubricants and reduce broach friction and wear, as shown in Figure 13b; secondly, since the fluff has the ability of 100% elastic recovery, it not only can sweep the microchip, but also act as a buffer effect when the fluff returns to its original position, as shown in Figure 13c. Finally, because the fluff has a certain height, it can also play a role in storing chips in the broaching process. When the fluff is bent, the microchips can be preserved in the interior, as shown in Figure 13d. Therefore, the purpose of reducing the furrow of the chip surface is achieved.

The reason for the serious furrow of FB1 is that there are too many microchips stored on the rake face of the broach, but they cannot be swept out in time, resulting in the fluff being unable to wrap the microchip, which induces serious friction between the tool and the chip. The larger distribution interval of fluff area on FB3 and FB4 results in excessive loss of cutting fluid and more furrows. FB2 can avoid this phenomenon and play the best role as a flocked broach in the broaching experiment, which is attributed to the existence of a reasonable flocking area.

### 4.3. Influence of Different Tools on Surface Quality

#### 4.3.1. Surface Quality of the Workpiece 

It can be seen from Figure 14 that with the increase in the cutting speed, the surface roughness of the workpiece decreases significantly, and the surface quality improves. At different broaching speeds, the FB show better surface roughness compared with the NB. In addition, FB2 shows relatively the best surface quality, and the NB surface quality is the worst. At the speed of 40 mm/s, the surface roughness of FB1–FB4 is reduced by 17%, 38%, 31%, and 14%, respectively, compared with the NB, which indicates that the FB contribute to improving the surface quality of the workpiece.

Figure 15 shows the surface morphology of the workpiece by the FB and the NB at the cutting speed of 40 mm/s. It can be seen that the NB has obvious chatter and ablation, which indicate that the wear mechanism is abrasion and serious vibrations are happening. In addition, chatter and ablation can be observed on FB1, but this phenomenon is slightly better in comparison to the NB. Furthermore, FB3 and FB4 also have different degrees of chatter and ablation. The wear of FB2 is not obvious, and only slight ablation exists, caused by the adhesion of microchips, thus indicating that a suitable flocking area and spacing can efficiently improve the surface quality of the workpiece.

This is because the microchips are subjected to friction and extrusion at the tool–chip interface as the cutting edge passes through the workpiece surface, so the temperature of the tool–chip interface becomes higher, resulting in the amount of ablation on the machined surface. More importantly, the machined surface produces many chatters due to vibration. Therefore, when periodic excitation occurs, the broaching system will exist with exciting vibration [36].

#### 4.3.2. Broaching Vibration of Different Tool Surfaces

In order to evaluate the vibration suppression effect of broaches flocked with fluff, the time-domain data of the cutting force (*F_c_*) of five kinds of broaches were converted into the frequency domain for analysis by the Fourier transform formula at the cutting speed of 40 mm/s, and the time window of 1.57 s. As illustrated in Figure 16, the peaks appear due to the experiment system producing vibration induced by the cutting force and the positive force when the broach passes through the workpiece. Small peaks can be observed around the broach passing frequency. These small peaks are caused by broach jump and chip thickness changes [37]. The maximum average amplitude appears in the NB, which indicates that the NB has serious vibration during machining. However, compared with the NB, the average amplitude of the FB is relatively small. Therefore, it can be explained that the flocked broach can play a beneficial role in suppressing vibration. The average amplitude fluctuation of FB2 is the smallest.

The deformation caused by chatter and the thermal deformation of the workpiece caused by cutting temperature can increase the average friction coefficient on the tool–workpiece interface [38], which will affect the surface quality of the workpiece, thereby reducing the surface roughness [39]. In addition, due to the coupling chatter in the machining process, there are obvious chatter and ablation on the surface of the workpiece [40,41]. The nylon material on the broach surface has elastic recovery ability, so it can suppress the vibration [4,42], which results in the surface quality of the workpiece being closer to the ideal surface, as shown in Figure 17. For FB1, the chip cannot be swept in time due to the flocked area being too large, so the microchips pile up on the rake face, resulting in chatter similar to the NB. The storage capacity of lubricant for FB3 and FB4 is relatively poor due to their large flocking spacing, which leads to a poorer processing quality than FB1. Owing to the balanced distribution of fluff, FB2 has an excellent ability to improve the stability of broaching.

## 5. Conclusions

A new type of tool surface is proposed in this paper. A flocked tool imitating a maple leaf was created and the broaching performance of the tool with different surfaces has been evaluated. The results show that flocked fluff on the rake face effectively reduces the cutting force and friction coefficient, promotes chip curling, and improves machined quality. The findings are as follows:(1)Flocked broaches can effectively reduce the cutting force (*F_c_*), positive pressure (*F_t_*), and coefficient of friction (COF) due to their superior lubrication ability, when compared to the unflocked broach (NB). The maximum reductions are 16.82%, 24.41%, and 6.07%. Furthermore, a modified analytical model with less than 5.6% error indicates that by combining the flocking area with lubricant, the COF can be predicted more accurately.(2)Because the fluff on the rake face improves lubricant absorption and the ability to store and sweep microchips, flocked broaches can improve chip curling and increase the shear angle, making it easier to achieve the purpose of chip breaking than with an NB, with a maximum increase in the shear angle of approximately 6.92%.(3)Flocked broaches can significantly improve machined quality and reduce surface roughness. The reason for this is that they can suppress the excitation vibration generated during the broaching process, improving chip formation stability and continuity.(4)FB2 has the best broaching performance of any broach due to its appropriate flocking area and spacing. The coating sizes of flocked surfaces below or above this threshold will cause a deterioration of the broaching performance.

## Figures and Tables

**Figure 1 materials-15-09040-f001:**
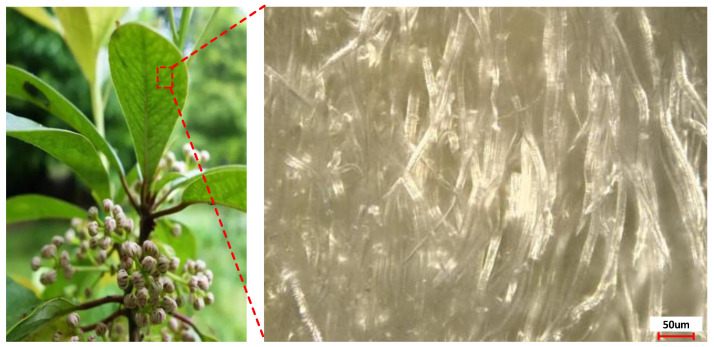
Back micromorphology of DCB leaves.

**Figure 2 materials-15-09040-f002:**
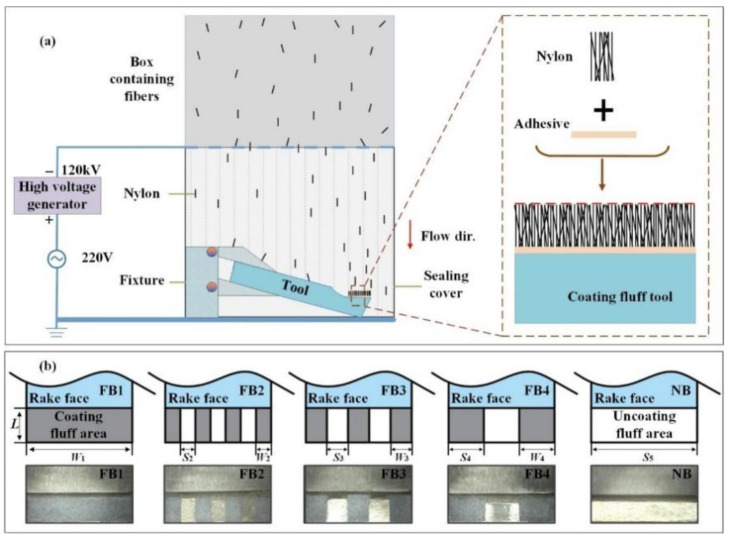
(**a**) Principle of electrostatic flocking; (**b**) Design styles and surface morphology of five broaches.

**Figure 3 materials-15-09040-f003:**
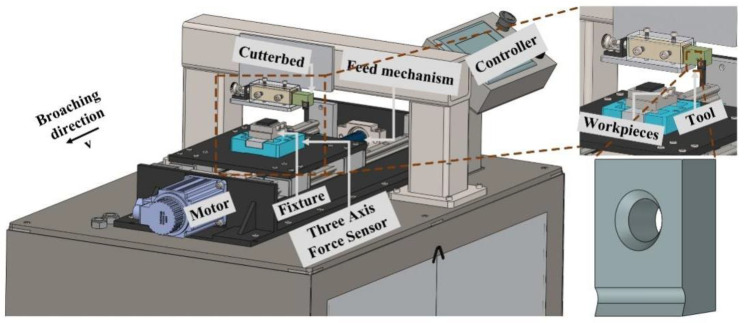
Broaching test site layout.

**Figure 4 materials-15-09040-f004:**
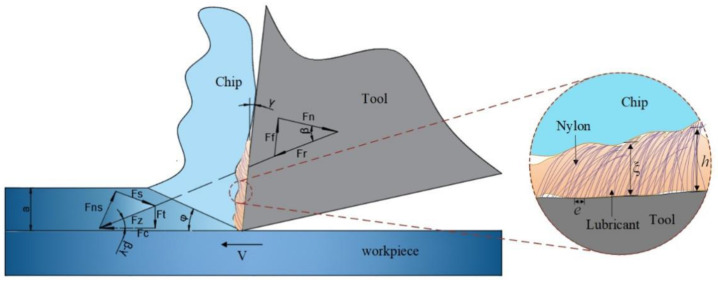
Two-dimensional equilibrium force system of the flocked broaches.

**Figure 5 materials-15-09040-f005:**
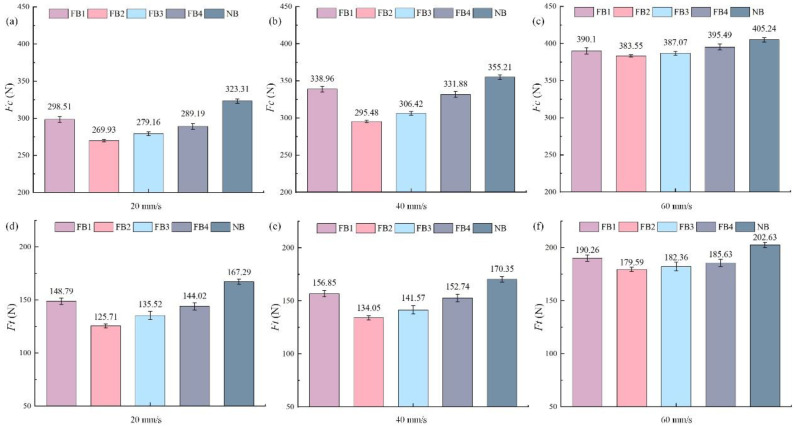
Broaching force by different tool surfaces at three cutting speeds: (**a**–**d**) cutting force (*F_c_*) at 20–60 mm/s.; (**d**–**f**) positive pressure (*F_t_*) at 20–60 mm/s.

**Figure 6 materials-15-09040-f006:**
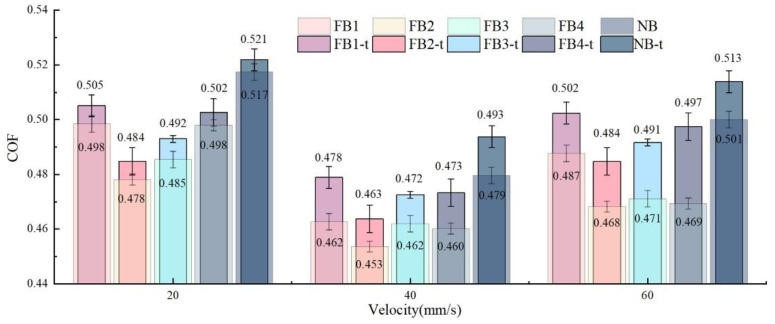
Comparison of theoretical COF (FB1–FB4) and actual COF (FB1-t–FB4-t) of broaches with different surfaces.

**Figure 7 materials-15-09040-f007:**
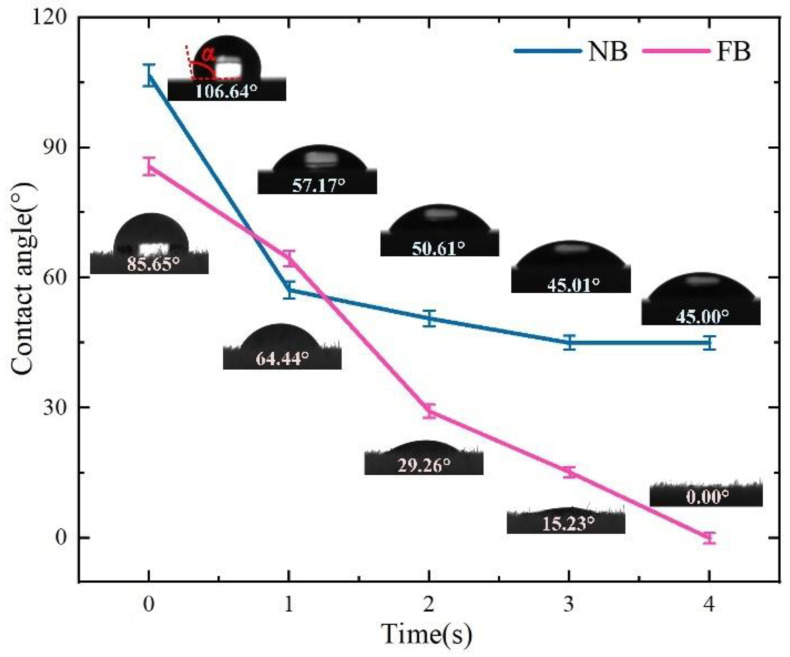
Contact angle of the rake face on the flocked broaches and ordinary broaches.

**Figure 8 materials-15-09040-f008:**
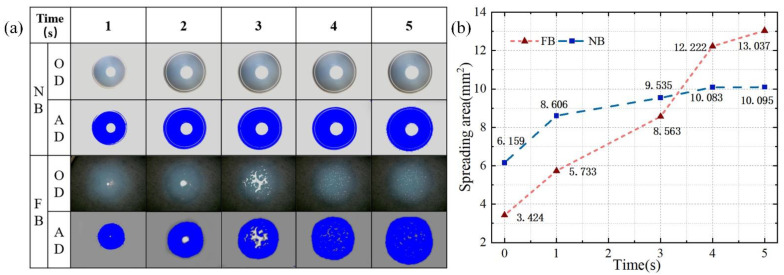
(**a**) Spreading experiment of an oil droplet on the tool surface; (**b**) Comparison of the spreading of an oil droplet on the tool surface of the FB and the NB.

**Figure 9 materials-15-09040-f009:**
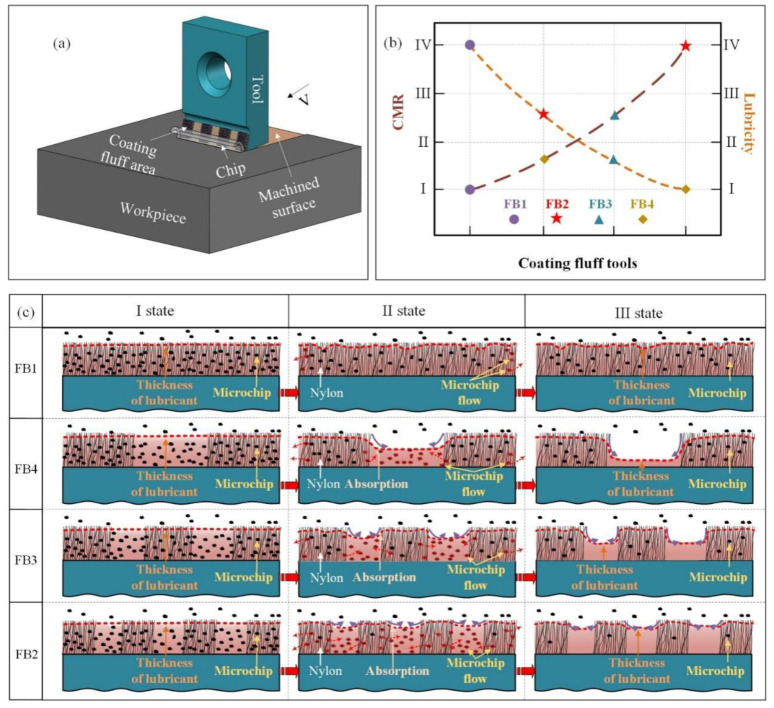
(**a**) Broaching processing flocking tools; (**b**)The CMR and lubricity of flocking tools; (**c**) Broaching mechanism of flocking tools.

**Figure 10 materials-15-09040-f010:**
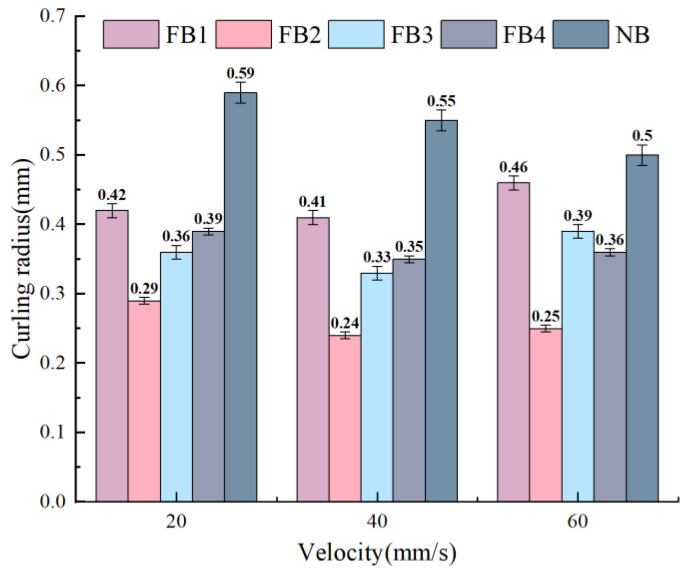
Chip curl radius of broaches with different surfaces.

**Figure 11 materials-15-09040-f011:**
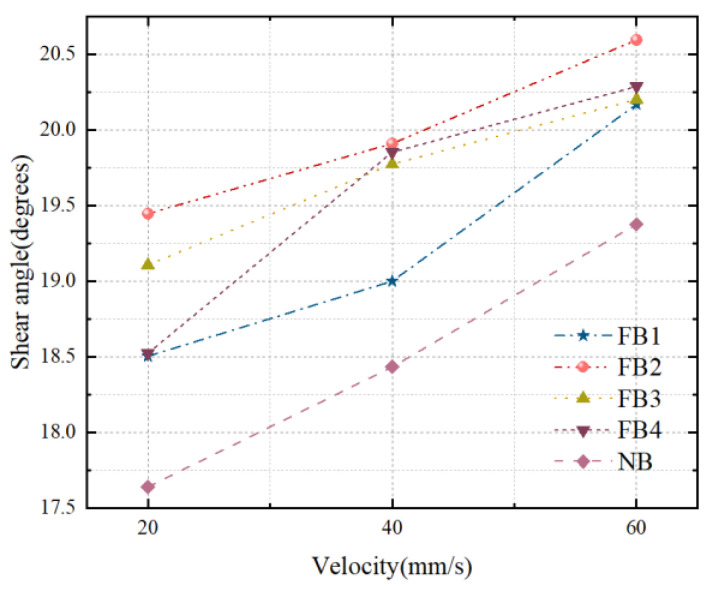
Cutting angle of broaches with different surfaces.

**Figure 12 materials-15-09040-f012:**
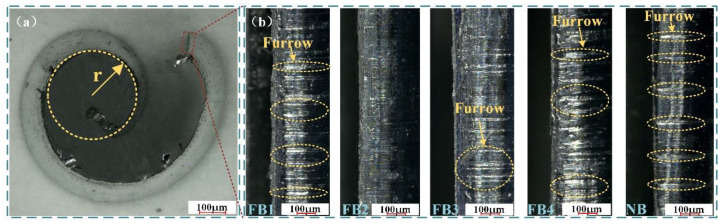
(**a**) Morphology of chip curl; (**b**) The end of chip morphology by different broach surfaces at 40 mm/s.

**Figure 13 materials-15-09040-f013:**
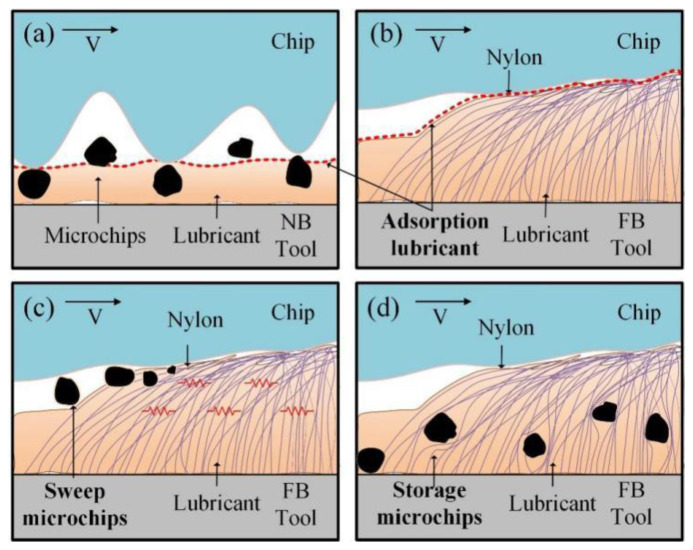
(**a**) Formation mechanism of chip morphology by the NB; (**b**) Adsorption capacity of the flocked broach for lubricant; (**c**) Effect of elastic recovery of fluff on sweeping microchip; (**d**) The function of the flocked broach to store microchips.

**Figure 14 materials-15-09040-f014:**
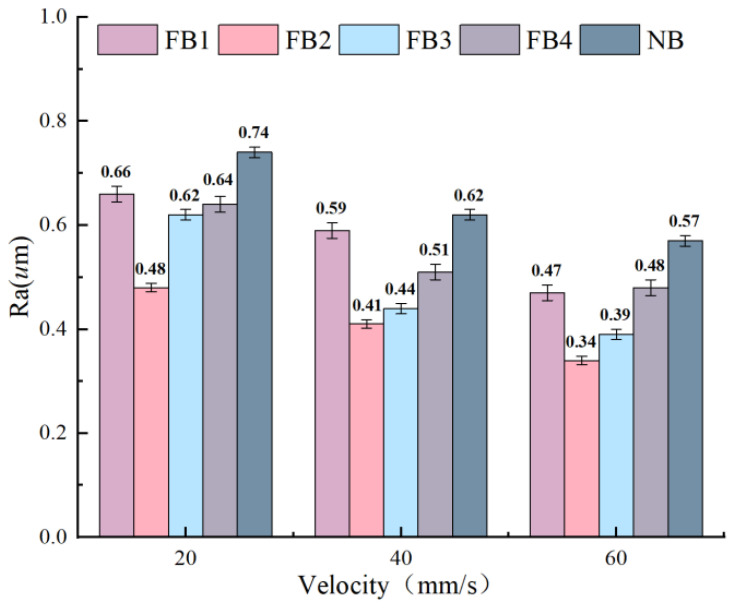
Roughness of the machined surface at different cutting speeds.

**Figure 15 materials-15-09040-f015:**
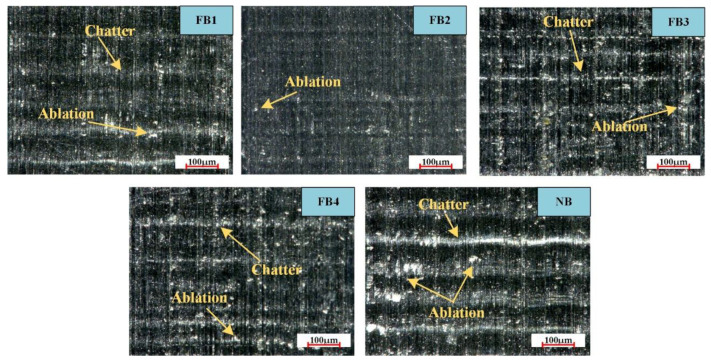
Surface morphology of the workpieces by different broach surfaces at 40 mm/s.

**Figure 16 materials-15-09040-f016:**
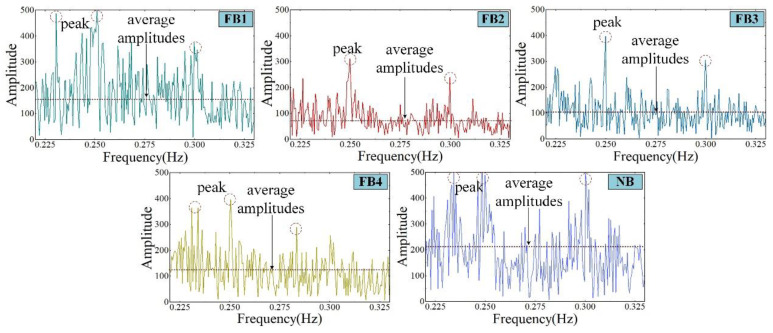
Broaching amplitudes of different tools at 40 mm/s.

**Figure 17 materials-15-09040-f017:**
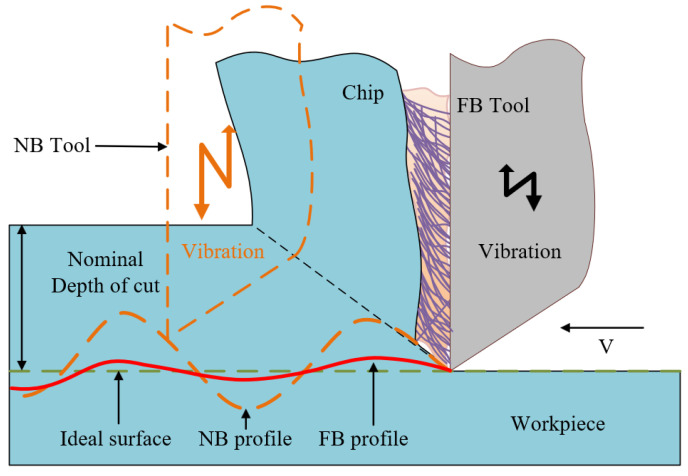
Vibration waveform of broaches with different surfaces.

**Table 1 materials-15-09040-t001:** Mechanical properties of the YG8.

Density	Impact Energy	Hardness	Bending Strength
g/cm^3^	J/cm^2^	HRA	MPa
14.6–14.9	2.5	89	1500

**Table 2 materials-15-09040-t002:** Composition of 7075 aluminum alloy.

Element	Cr	Fe	Si	Mn	Cu	Mg	Zn	Ti	Other	Al
wt%	0.18–0.28	0.5	23.6	0.3	1.2–2.0	2.1–2.9	5.1–6.1	0.2	0.15	bal

**Table 3 materials-15-09040-t003:** Properties of 7075 aluminum alloy at normal temperature.

Yield Strength	Tensile Strength	Elongation Ratio	Density	Hardness	Melting Point
$\;\;\;\;\sigma_{s}\;(\mathrm{MPa})$	$\sigma_{v}\;(\mathrm{MPa})$	δ (%)	ρ (g/cm^3^)	(HRC)	℃
503	572	11	2.82	150	475–635

**Table 4 materials-15-09040-t004:** The parameters of electrostatic flocking.

Voltage	Time	Distance	Fiber Length	Fiber Diameter
(V)	(s)	(mm)	(mm)	(µm)
120	600	30	0.8	8

**Table 5 materials-15-09040-t005:** Types and geometric parameters of broach flocking.

Tool Type	Size of Geometries (mm)		
	*L*	$w_{i}$	$s_{i}$
FB1	3	13	0
FB2	3	1.86	1.86
FB3	3	2.6	2.6
FB4	3	4.33	4.33
NB	3	0	0

**Table 6 materials-15-09040-t006:** Parameters of the broaching experiment.

Broaching Speed	Broaching Depth	Broaching Stroke
(mm/s)	(mm)	(mm)
20	0.05	40
40	0.05	40
60	0.05	40

## Data Availability

Not applicable.

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
