# Peer review of "Assessment of a Bionic Broach Implanted with Nylon Fibers"

_materials, 2022, doi:10.3390/ma15249040_

Round 1

Reviewer 1 Report

Dear Authors,

the paper is well written. 

I suggest the revision of figures no. 5, 6 and 7.

I recommend the following bibliography to give completeness to the treatment and conclusions:

1 - Fabbrocino F., Carpentieri G., Three-dimensional modeling of the wave dynamics of tensegrity lattices, COMPOSITE STRUCTURES (IF: 4.829), (2017), Vol. 173, pag. 9-16, ISSN: 0263-8223, DOI: 10.1016/j.compstruct.2017.03.102;

2 - Mancusi G., Fabbrocino F., Feo L., Fraternali F., Size effect and dynamic properties of 2D lattice materials, COMPOSITES. PART B, ENGINEERING (IF: 6.864), 2017, Vol. 112, pag. 235-242, ISSN: 1359-8368, DOI: 10.1016/j.compositesb.2016.12.026.

Best regards

Reviewer 2 Report

The article is nicely written and well proofread, except for a line break at the end of the introduction and minor typos in the conclusions.

Comments 

Why is the nylon chosen that does not have the lowest coefficient of friction and not the best mechanical properties. Perhaps PTFE fibers could be expected to give better results. This question arises because "nylon fibers" are mentioned in the title of the publication.

The size of polyamide fibers. Does it play a role? 

How does the model take into account the specific properties of polymer fibers?

How stable and resistant are the formed flocking surfaces to multiple mechanical loads.
